# ON THE CRITICAL ROLE OF CONVENTIONS IN ADAPTIVE HUMAN-AI COLLABORATION

**Andy Shih[†], Arjun Sawhney[†], Jovana Kondic[‡], Stefano Ermon[†] & Dorsa Sadigh[†]**
[†]Stanford University, [‡]Princeton University
[†]{andyshih,arjunsawhney,ermon,dorsa}@cs.stanford.edu
[‡]jkondic@princeton.edu

## ABSTRACT

Humans can quickly adapt to new partners in collaborative tasks (e.g. playing basketball), because they understand which fundamental skills of the task (e.g. how to dribble, how to shoot) carry over across new partners. Humans can also quickly adapt to similar tasks with the same partners by carrying over conventions that they have developed (e.g. raising hand signals pass the ball), without learning to coordinate from scratch. To collaborate seamlessly with humans, AI agents should adapt quickly to new partners and new tasks as well. However, current approaches have not attempted to distinguish between the complexities intrinsic to a task and the conventions used by a partner, and more generally there has been little focus on leveraging conventions for adapting to new settings. In this work, we propose a learning framework that teases apart *rule-dependent* representation from *convention-dependent* representation in a principled way. We show that, under some assumptions, our rule-dependent representation is a sufficient statistic of the distribution over best-response strategies across partners. Using this separation of representations, our agents are able to adapt quickly to new partners, and to coordinate with old partners on new tasks in a zero-shot manner. We experimentally validate our approach on three collaborative tasks varying in complexity: a contextual multi-armed bandit, a block placing task, and the card game Hanabi.

## 1 INTRODUCTION

Humans collaborate well together in complex tasks by adapting to each other through repeated interactions. What emerges from these repeated interactions is shared knowledge about the interaction history. We intuitively refer to this shared knowledge as *conventions*. Convention formation helps explain why teammates collaborate better than groups of strangers, and why friends develop lingo incomprehensible to outsiders. The notion of conventions has been studied in language (Clark & Wilkes-Gibbs, 1986; Clark, 1996; Hawkins et al., 2017; Khani et al., 2018) and also alluded to in more general multiagent collaborative tasks (Boutilier, 1996; Stone et al., 2010; Foerster et al., 2019; Carroll et al., 2019; Lerer & Peysakhovich, 2019; Hu et al., 2020). For example, Foerster et al. (2019) trained agents to play the card game Hanabi, and noted the emergent convention that "hinting for red or yellow indicates that the newest card of the other player is playable".

One established characterization of a convention that is commonly used (Boutilier, 1996; Hawkins et al., 2017; Lerer & Peysakhovich, 2019) is an *arbitrary* solution to a recurring coordination problem (Lewis, 1969). A convention is thus one of many possible solutions that a group of partners happens to converge to. This is in contrast to problem solutions that are enforced by the rule constraints, and would have arisen no matter how the partners collaborated and what behavior they converged to. Success in a collaborative task typically involves learning both types of knowledge, which we will refer to as *convention-dependent* and *rule-dependent* behavior. The distinction between these two types of behavior has been studied extensively in the linguistic literature (Franke et al.; Brochhagen, 2020). In this work, we focus on the less-explored setting of implicit communication, and provide a concrete approach to learn representations for these two different types of behavior.

In the context of multi-agent or human-AI collaboration, we would like our AI agents to adapt quickly to new partners. AI agents should be able to flexibly adjust their partner-specific convention-

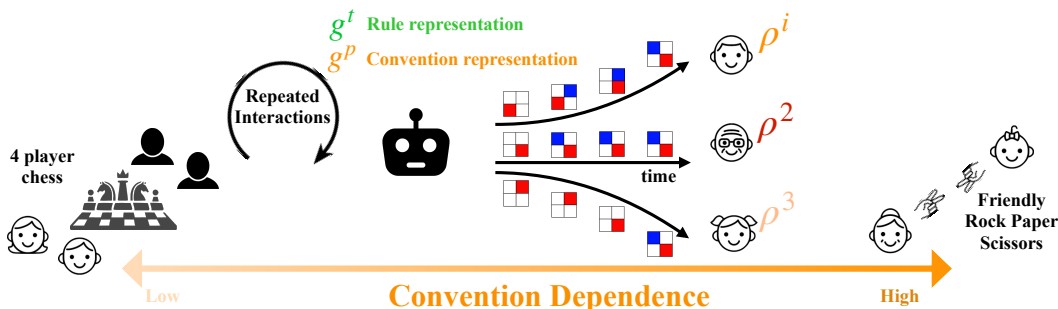

Figure 1: Partners form conventions in a collaborative task through repeated interactions. An AI agent can adapt quickly to conventions with new partners by reusing a shared rule representation $g^t$ and learning a new partner-specific convention representation $g^p$. Certain collaborative tasks, such as friendly Rock-Paper-Scissors, are more convention-dependent than others, such as 4-player chess.

dependent behavior while reusing the same rule-dependent behavior to simplify the learning problem – just like how humans quickly adapt when playing basketball with a new friend, without the need to re-learn the rules of the game. We would also like our AI agents to coordinate well on similar tasks when paired with the same partners – just like how humans can coordinate better when playing a new sport but with an old friend.

Although many existing works similarly recognize conventions as an important factor of successful collaboration, they do not focus on separating conventions from rules of the task. This means that adapting to a new partner can be as difficult as learning a new task altogether. For example, existing techniques that emphasize modeling the partner's policy, such as theory of mind (Simon, 1995; Baker et al., 2017; Brooks & Szafir, 2019) or multi-agent reinforcement learning (Foerster et al., 2018), attempt to model *everything* about the agent's belief of the partner's state and policies. Such belief modeling approaches very quickly become computationally intractable, as opposed to solely focusing on the relevant conventions developed with a partner.

To address the challenges above, we propose a framework that explicitly separates convention-dependent representations and rule-dependent representations through repeated interactions with multiple partners. After many rounds of solving a task (e.g. playing basketball) with different partners, an AI agent can learn to distinguish between conventions formed with a specific partner (e.g. pointing down signals bounce pass) and intrinsic complexities of the task (e.g. dribbling). This enables us to leverage the representations separately for fast adaptation to new interactive settings.

In the rest of this paper, we formalize the problem setting, and describe the underlying model of partners and tasks. Next, we present our framework for learning separations between rule and convention representations. We show that, under some conditions, our rule representation learns a sufficient statistic of the distribution over best-response strategies. We then run a study on human-human interactions to test if our hypothesis – that partners can carry over the same conventions across tasks – indeed holds for human subjects. Finally, we show the merits of our method on 3 collaborative tasks: contextual bandit, block placing, and a small scale version of Hanabi (Bard et al., 2020).

## 2 RELATED WORK

Convention formation has been studied under the form of iterated reference games (Hawkins et al., 2017; 2019), and language emergence (Mordatch & Abbeel, 2018; Lazaridou et al., 2017). In these works, partners learn to reference objects more efficiently by forming conventions through repeated interactions. But in these tasks, there is little intrinsic task difficulty beyond breaking symmetries.

In multi-agent reinforcement learning, techniques such as self-play, cross-training, or opponent learning (Foerster et al., 2018) have been used to observe emergent behavior in complex, physical settings (Nikolaidis & Shah, 2013; Liu et al., 2019; Baker et al., 2020). In addition, convention formation has been shown in tasks like Hanabi (Foerster et al., 2019; Hu & Foerster, 2019), Overcooked (Carroll et al., 2019; Wang et al., 2020), and negotiation tasks (Cao et al., 2018), where agents learn both how to solve a task and how to coordinate with a partner. But, these works qualitatively analyze emergent conventions through post-hoc inspection of the learned policies, and do not learn representations

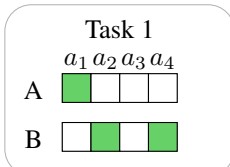 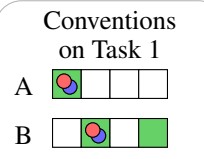 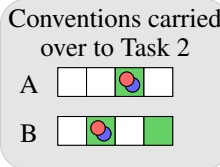

Figure 2: Collaborative contextual bandit task. On the left we see a task with 2 contexts represented by rows and 4 arms represented by cells. At each context, the prize if both partners pull the same green arm is 1, and 0 otherwise. In the middle, we see a possible convention that two partners may converge to: pulling $a_1$ in context A, and pulling $a_2$ in context B. On the right, when the task changes in context A but stays the same in context B, we expect the same two partners to continue to pull $a_2$ in context B.

that separate conventions from rule-dependent behavior. This makes it difficult to leverage learned conventions when adapting to new partners. More recently, an approach was proposed to design agents that avoid relying on conventions altogether; instead, the agents aim for unambiguous but potentially sub-optimal strategies (Hu et al., 2020).

Other approaches for studying conventions include game theory (Lerer & Peysakhovich, 2019) and theory of mind (Simon, 1995; Rabinowitz et al., 2018). Pragmatic reasoning (Goodman & Frank, 2016) is another framework that aims to estimate beliefs over the partner's states and policies, but all these frameworks that rely on modelling beliefs quickly become intractable. Instead of approximating these belief modelling frameworks (Monroe & Potts, 2015), we instead focus on learning representations for rules and conventions. Our work shares similarities with the paradigms of meta-learning (Schmidhuber, 1987; Finn et al., 2017) or multi-task learning (Caruana, 1997), if we view collaboration with different partners as new but related tasks. However, we are also interested in how conventions persist across tasks with similar symmetries.

More related to our work is perhaps modular policy networks (Devin et al., 2017), which learns policy modules and composes them together to transfer across robot arms with different degrees of freedom and across different tasks. However, their approach relies on input cleanly split between background observations of the task, and robot's internal observations. We do not assume such a split in the input; our approach tries to learn the separation between rules and partners with one input channel.

## 3 PRELIMINARIES

We begin by formally defining our problem setting as a two-player Markov Decision Process (MDP).

**Two-Player MDP** We consider a two-agent MDP with identical payoffs, which is a tuple $\{S, \{A_e, A_p\}, P, R\}$. $S$ is a set of states, $A_e$ is a set of actions for agent $e$, $A_p$ is a set of actions for agent $p$. In general, $e$ represents the ego agent (that we control), and $p$ represents agents who partner with $e$. $P : S \times A_e \times A_p \times S \to [0, 1]$ is the probability of reaching a state given the current state and actions of all agents, and $R : S \times S \to \mathbb{R}$ is the real-valued reward function. Since we are interested in repeated interactions, we consider finite-horizon MDPs. A policy $\pi$ is a stochastic mapping from a state to an action. For our setting, our policy $\pi = (\pi_e, \pi_p)$ has two parts: $\pi_e : S \to A_e$ and $\pi_p : S \to A_p$, mapping the state into actions for agent $e$ and $p$ respectively.

We also consider collaborative tasks that are partially observable in our experiments. In addition, some tasks are turn-based, which can be handled by including the turn information as part of the state, and ignoring the other player's actions. Here, we focus on tasks with discrete state and action spaces.

**Running Example: Collaborative Contextual Bandit** We describe a collaborative task to ground our discussion around adapting to new partners, and adapting to new tasks with old partners. Consider a contextual bandit setting with 2 contexts, 4 actions, and a prize for each action. The two players each independently pick an arm to pull, and they score prize($a$) points if they both picked the same arm $a$; otherwise, they score 0 points. An example is depicted in Figure 2, where green boxes represent arms with prize 1, the rest have prize 0, and red and blue circles show the player's actions.

In Task 1 of Figure 2, context A only has one good arm $a_1$ while the context B has two good arms $a_2$ and $a_4$. After repeated rounds of playing, two agents can eventually converge to coordinating on a convention that chooses one of the two good arms in context B (e.g. selecting the leftmost good arm $a_2$). When the task shifts slightly as shown in Task 2 but context B remains the same across the two

tasks, we can reasonably expect the partners to adhere to the same convention they developed for context B of Task 1 when playing context B of Task 2.

There are two axes of generalization at play. First, for a fixed task we would like to learn the underlying structure of the task (e.g. green arm locations) to quickly adapt to a new partner when playing the same task. Second, for a fixed partner we would like to keep track of developed conventions to quickly coordinate with this partner on a new task. In the next sections, we define the notion of a new task and a new partner, and propose a framework to effectively generalize across both dimensions.

# 4 MODEL OF PARTNERS AND TASKS

We consider a family of tasks that share the same state space, action space, and transition dynamics (all of which we refer to as the domain), but can differ in the reward function of the underlying MDP. We also consider a family of partners that our ego agent would like to coordinate with. The underlying model governing the actions taken by a partner at a state of a given task can be seen in Figure 3. For a new task, a new reward function is sampled from $\theta_R$, while the domain of the MDP is fixed. For a new partner, we sample from $\theta_\rho$ a new function $\rho$, which maps the $Q$-function at a state of the task to an action. We refer to $\rho$ as the convention of a partner. In other words, the action $a_s^{it}$ of a partner $i$ at state $s$ of task $t$ depends on the partner's conventions and (indirectly through the $Q$-function at state $s$) the reward of the task. $\theta_\rho$ is the distribution over conventions and can correspond to, for example, the distribution over emergent conventions from AI-agents trained from self-play, or can be derived from human partners.

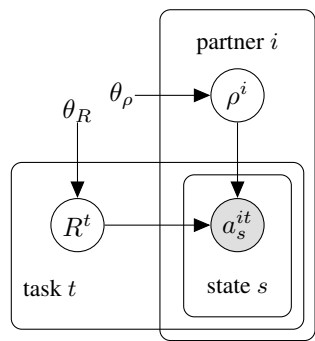

Figure 3: Generating process of partner's actions for a two-player MDP. For each task we first sample a reward function $R^t$ for the MDP. This determines the $Q$-function at each state. For each partner $i$ at each state $s$ of task $t$, we sample a function $\rho^i$ that maps the $Q$-function at state $s$ to an action $a_s^{it}$.

More formally, for a fixed domain let $Q_s^R(a_p) = \max_{a_e} Q^R(s, a_e, a_p)$, where $Q^R$ is the optimal $Q$-function for the two player MDP with reward $R$ (Boutilier, 1996; Oliehoek et al., 2008). In other words, $Q_s^R$ is the $Q$-function from the partner's perspective at state $s$ in the task with reward $R$ assuming best response by the ego agent. Then, the convention of a partner $\rho : S \times \mathbb{R}^{|A_p|} \to A_p$ determines the partner's action at state $s$, given by $\rho^i(s, Q_s^R)$. For example, we might expect the distribution of actions across different partners at a state $s$ to follow Boltzmann rationality (Ziebart et al., 2008): $\mathbb{E}_{\rho \sim \theta_\rho}[\mathbb{1}[\rho(s, Q_s^R) = a_p]] \propto \exp(Q_s^R(a_p))$. Lastly, we assume that behavior at different states are uncorrelated: a choice of action at one state tells us nothing about actions at other states.

Given this formulation, we can hope to learn the underlying structure of a task by playing with a number of partners on the same task. For example, if many sampled partners all make the same action $a_p$ at a state $s$, it is likely that $Q_s^R(a_p)$ is much higher than the second best action, i.e., the optimal action is dependent on the rules of the task as opposed to the arbitrariness of partner strategies. Therefore a new partner will likely take action $a_p$ as well. Additionally, when coordinating with the same partner across different tasks, we can expect to see developed conventions persist across states with similar $Q$-functions. In particular, if $Q_s^{R1} = Q_s^{R2}$ for two different tasks with reward functions $R1$ and $R2$, then $\rho^i(s, Q_s^{R1}) = \rho^i(s, Q_s^{R2})$, so a partner will take the same action at state $s$ across the two tasks (e.g. context B in Figure 2 across Task 1 and 2).

We note that the roles of partners and tasks in our model are asymmetrical, assuming rational partners. A task can completely determine the actions of all partners at states with one good action (e.g. context A in Figure 2), whereas a partner cannot blindly pick the same action across all tasks. This asymmetry is reflected in our learning framework and our setup of adapting to new partners and new tasks.

# 5 LEARNING RULES AND CONVENTIONS

With the goal of adapting to new partners and tasks in mind, we propose a framework aimed at separating the rule representations associated with a task from the convention representations

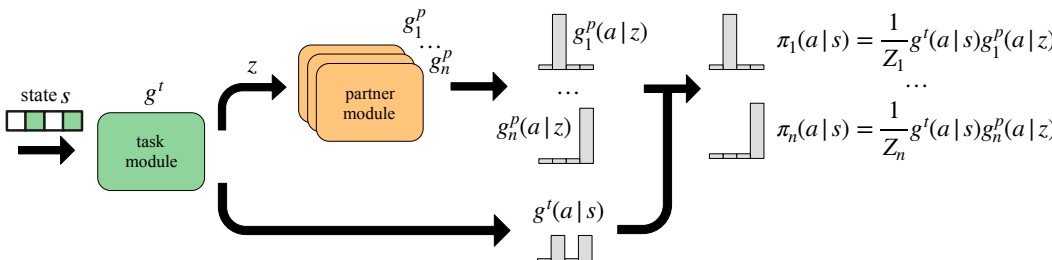

Figure 4: Policy network with separate task/partner modules to separate rule/convention representations. The task module $g^t$ maps the state observations to an action distribution $g^t(a|s)$ and to latent variables $z$. Each partner module maps $z$ to another action distribution $g^p_i(a|z)$. The policy of the ego agent when playing with partner $i$ is set to be proportional to the product of $g^t(a|s)$ and $g^p_i(a|z)$.

associated with a partner. At training time, we assume access to a task and a set of partners sampled from the same partner distribution $\theta_\rho$. When adapting to a new partner sampled from $\theta_\rho$, we would like to learn a new convention representation but keep the learned rule representation. When adapting to a new task with the original partners, we would like the opposite: to learn a new rule representation but keep the learned convention representation. In our setup, the ego agent knows the identity of the tasks and partners, and the goal is to maximize the total reward with a new task and partner (the combination of which may be new to the ego agent). Naturally, this points us towards a modular architecture that enables integrating new combinations of partner and task representations.

We propose the architecture shown in Figure 4, where we learn to play with each of the training partners via reinforcement learning. Each rectangular module represents a multi-layer perceptron, and each box with grey bars represents a distribution over actions in the action space of the MDP. When training with multiple partners on the same task, the policy network uses one single shared task module $g^t$, and uses one partner module $g^p_i$ for each partner $i$. The task module takes the state observations $s$ as input, and outputs latent variables $z$ and an action distribution $g^t(a|s)$. Then, each partner module takes $z$ as input, and outputs an action distribution $g^p_i(a|z)$. Here, $g^p_i(a|z)$ does not represent partner $i$'s action distribution, but rather represents our ego agent's action distribution in response to partner $i$. The policy $\pi_i$ of the policy network for partner $i$ is set to be proportional to the product of the action distributions.

$$\pi_i(a|s) = \frac{1}{Z_i} g^t(a|s) \cdot g^p_i(a|z)$$

As it stands, many possible combinations of task and partner module outputs could lead to optimal action distributions $\pi_i(a|s)\,\forall i$. To encourage the task/partner modules to learn the rule/convention representations respectively, we include a regularization term that pushes $g^t(a|s)$ to be the marginal best-response strategy over partners at state $s$. In practice, this amounts to minimizing the Wasserstein distance between the task module's output and the average best-response strategies $\pi_i$.

$$D(s) = \sum_{a \in A} \left| g^t(a|s) - \frac{1}{n} \sum_i \pi_i(a|s) \right|$$

By pushing the task module $g^t$ to learn the marginal best-response strategy, we can cleanly capture the rule-dependent representations of the task, to be used when adapting to a new partner. In fact, under some assumptions, we can show that $g^t$ is learning the optimal representation, as it is a sufficient statistic of the distribution over best-response strategies across possible partners.

In particular, for a fixed task let $f(s, a_e, a_p)$ represent the probability of the ego agent taking action $a_e$ using its best-response strategy to partner action $a_p$ at state $s$. So, $\sum_{a_e} f(s, a_e, a_p) = 1$. We say the ego agent is deterministic if it can only take on deterministic strategies: $\forall s, a_p \, \exists \, a_e : f(s, a_e, a_p) = 1$.

**Lemma 1.** *For a deterministic ego agent, the marginal best-response strategy across partners is a sufficient statistic of the distribution of best-response strategies across partners.*

Next, we lay out the learning procedure in more detail in Algorithm 1. We use the task module $g^t$ and the partner module $g^p_i$ when playing with partner $i$. We push the task module to learn the marginal best-response action by adding the loss term $D$ on top of a standard policy gradient (PG) loss term.

---

**Algorithm 1:** Learning Separate Representations for Partners and Tasks

---

**Input:** A MDP $M$, $n$ partners $\rho_1 \ldots \rho_n \sim \theta_\rho$
**Output:** Policy network modules for adapting to new partners from $\theta_\rho$ and new tasks

  **1** Initialize the task module $g^t$ and $n$ partner modules $g_1^p, \ldots, g_n^p$
  **2** $G, T \leftarrow \{g^t, g_1^p, \ldots, g_n^p\}$, number of iterations
  **3 for** $j \leftarrow 1, T$ **do**
  **4**     **for** $i \leftarrow 1, n$ **do**
  **5**         Collect rollouts $\tau = (\boldsymbol{s}, \boldsymbol{a}, \boldsymbol{r})$ using $\{g^t, g_i^p\}$ with partner $\rho_i$ on $M$
  **6**         Update $G$ with loss $L^{\mathrm{PG}}(g^t, g_i^p, \tau) + \lambda \mathbb{E}_{\boldsymbol{s}}[D(s)]$
**Return:** $G$

---

**Adapting to new partner** When adapting to a new partner, we reuse the task module $g^t$ along with a reinitialized partner module, and train both modules. At states with only one optimal action $a^\star$ regardless of the partner's convention, the marginal best-response strategy will concentrate on $a^\star$. As such, the task module $g^t$ will immediately push the ego agent to choose $a^\star$, improving adaptation time. In contrast, at states with many possible optimal actions depending on the partner's conventions, the marginal best-response strategy will spread the action distribution among the possible optimal actions, allowing the ego agent to explore amongst the optimal options with the new partner.

**Coordinating on a new task** When faced with a new task, we would like to recognize the parts of the task where conventions with old partners on an old task carry over. In particular, in states for which the joint $Q$-function has not changed, our partners models will take the same actions as before, and our ego agent can employ the same response as before. With this in mind, suppose we have trained a task module $g^t$ and a set of $2n$ partner modules $g_1^p, \ldots, g_{2n}^p$ with a set of partners on an old task. We can learn to coordinate on a new task with partners $1$ to $n$ by fine-tuning the task module $g^t$, paired with partner modules $g_1^p, \ldots, g_n^p$ with frozen parameters. At states where the joint $Q$-function has not changed, the same latent variable outputs $z$ will work well for all partners, so the task module will output the same $z$ and the actions of the ego agent will not change. Then, we can coordinate with partners $n+1$ to $2n$ in a zero-shot manner by pairing $g^t$ with partner modules $g_{n+1}^p, \ldots, g_{2n}^p$.

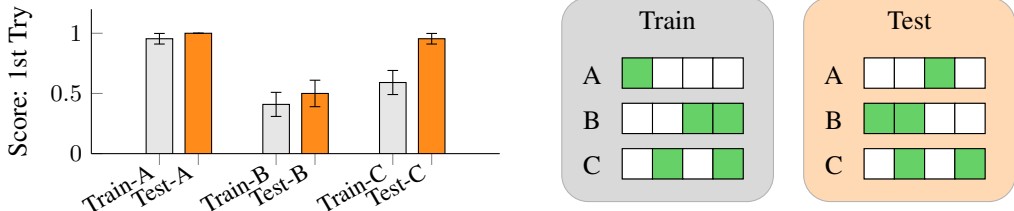

Figure 5: User study on the collaborative contextual bandit task. Participants were asked to coordinate on the Train task and, immediately afterwards, with the same partner on the Test task (shown on the right). Some contexts have two equally optimal actions (green boxes), requiring symmetry breaking. On the left we plot the score, averaged across participants, of the first try at each context. Our participants were able to coordinate well on Test-C on the first try, even though the context requires symmetry breaking.

## 5.1 UNDERSTANDING CONVENTIONS

When adapting to a new task with the same partner, we are relying on the hypothesis that our partners will carry over the same conventions developed on an earlier task, at states where the $Q$-function has not changed. Here, we would like to test this hypothesis and study if and when conventions persist across tasks, through a study of human-human interactions in the collaborative contextual bandit task.

**User Study: Conventions in Human-Human Interactions.** We conducted a study with 23 pairs of human partners (46 subjects), using the collaborative contextual bandit task from Section 3. The study was conducted via an online interface, with partners in separate rooms. The actions of partners are revealed to each other at the end of each try, and the partners had no other form of communication.

- *Independent Variables:* We vary the prize of the arms at different contexts, i.e., which arms are good to coordinate on. Pairs of partners were given 5 tries to coordinate on 3 different contexts (as shown

in Fig. 5). We then ask the same pair of partners to coordinate on the Test task with 3 new contexts.
- *Dependent Measures:* We measure the score of each pair's first try at each context. The maximum
score in each run for each context is 1, corresponding to the two partners picking the same good arm.
- *Hypothesis:* Our hypothesis is that partners can coordinate well on the test task by carrying over
over conventions developed on the train task.
- *Results:* On the left of Figure 5 we plot the average score of the first try (zero-shot) of our participants
at solving each context in each task. In the Train task (grey bars), the partner pairs are coordinating
for the first time. As expected, they generally had no trouble coordinating on Train-A (since there is
only 1 good option), but scored lower on Train-B and Train-C (since there are 2 good options each).

Immediately following the 5 trials on the training task, the partner pairs coordinated on each context
of the Test task (orange bars in Fig. 5), to see if they can leverage developed conventions on a new
task. On Test-A, they did well since there is only one good option. However, they coordinated much
better on Test-C compared to Test-B, even though both contexts have two good options. The main
difference between the two contexts is that Test-C shares the same set of optimal actions ($a_2$ and $a_4$)
as Train-C, whereas Test-B does not share the same set of optimal actions as Train-B.

Overall, this study suggests that human partners can successfully carry over conventions developed
through repeated interactions with their partners, and further coordinate better when carrying over
conventions across tasks at states where the set of optimal actions are similar (e.g. the higher gap for
same context such as context C across the train and test tasks).

## 6 EXPERIMENTS

We experiment with our approach on three coordination tasks: collaborative contextual bandit,
collaborative block placing, and Hanabi. We show results on adaptation to new partners for a fixed
task, and zero-shot coordination with same partners on a new task. We compare with 1) a baseline
(BaselineAgg) that learns a shared policy network to play with all training partners, using the average
of the gradients with each partner and 2) a baseline (BaselineModular) that similarly uses a modular
policy approach with separate modules for each partner, but does not explicitly represent the marginal
best-response strategy, and 3) First-Order MAML (Nichol et al., 2018). For our method, we vary $\lambda$
from $0.0$ to $0.5$: a higher value pushes the task module output to more closely match the marginal
action distribution. We also test the use of a low dimensional $z$ (the interface between the task and
partner module). In general, we find that our method with regularization performs the best, and that
it is unclear if using a low dimensional $z$ is helpful. The partners in our experiments were either
generated via self-play or were hand-designed to use varying conventions.

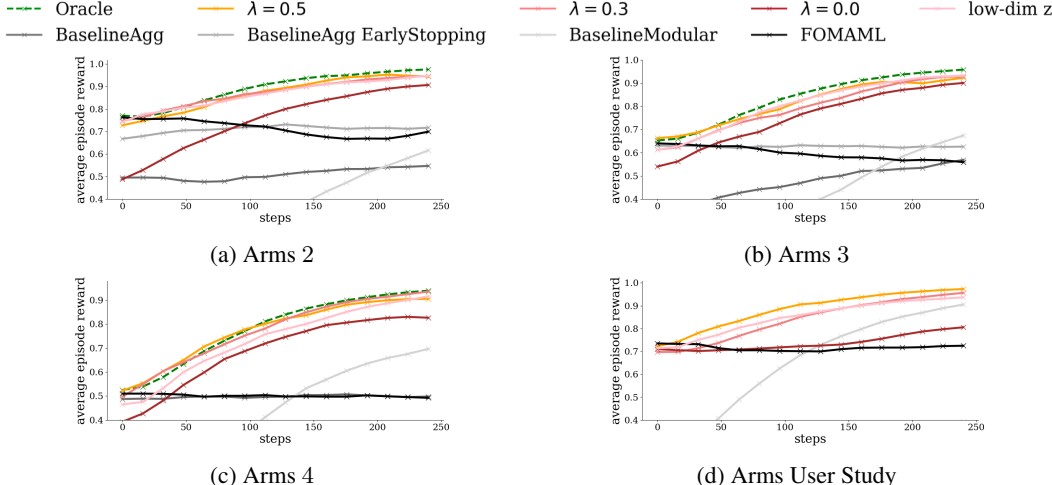

Figure 6: Contextual bandit game: adapting to a single new partner. In Figures 6a-c, we train and test on
hand-designed partners. "Arms $m$" refers to a task having $m$ contexts with symmetries (exact task details in the
Appendix). In Figure 6d, we train and test on partner policies derived from the data collected in our user study,
and we use the Train task shown in Figure 5.

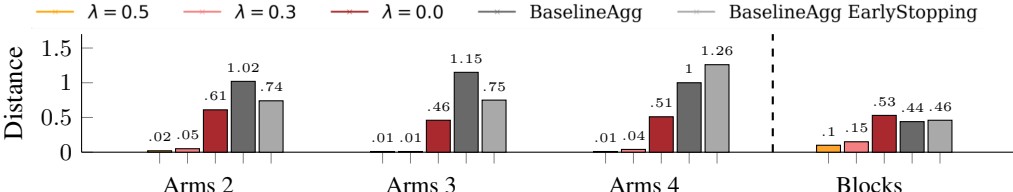

Figure 7: Wasserstein distance between task module output and true marginals assuming uniform preference over optimal actions. Lower is better. We do not compare with BaselineModular since it does not have output that can be interpreted as marginals. The left three sets of bars are for the contextual bandit task, and the right set of bars are for the blocks task. We omit Hanabi since it is not easy to determine which actions are optimal. It is interesting that even without regularization $\lambda = 0.0$, the distance to ground truth is still lower than the baselines, suggesting the model is learning some level of task-specific representation just due to the architecture.

**Contextual Bandit** We study the collaborative contextual bandit task described in Section 3, using 4 contexts and 8 arms. We study variations of the task by altering the number of contexts with more than one good option (i.e. symmetries). We write "Arms $m$" to refer to a task having $m$ contexts with symmetries – coordination should be easier for tasks having fewer contexts with symmetries.

In Figure 6a-c we show results on adaptation to new hand-designed partners (Figure 11 for self-play partners), varying the number of contexts with symmetries. We also plot the green oracle curve, by using our learning framework along with oracle best-response marginals (assuming uniform distribution over optimal actions across partners). To see if our method learns the correct marginals, we measure the Wasserstein distance between the learned marginals and the oracle marginals, in Figure 7. Using $\lambda = 0.5$ produces task module outputs that closely match the oracle marginals. To test zero-shot coordination with the same partners on new tasks, we tweak the task by altering contexts with only one good action, keeping the contexts with symmetries the same (similar to the change in Figure 2). We use hand-designed partners to make sure that partners use the same conventions at unchanged contexts. In Figure 10 we see that our method outperforms BaselineModular.

We include an additional plot in Figure 6d, where we instead use the Train task from the user study as shown in Figure 5. We use the data collected in our user study to create the partner policies, in lieu of the hand-designed partners. We observe similar trends – that the modular learning framework with marginal regularization performs the best.

**Block Placing** Next, we study a collaborative block placing task which involves a $2 \times 2$ goal-grid, with a single red block and a single blue block on the grid. The goal is to reconstruct the goal-grid from a blank working-grid, with partner 1 (P1) only controlling a red block and partner 2 (P2) only controlling a blue block. The partners take turns moving their blocks on the working-grid, and can see each other's actions on the working-grid. However, only P1 can see the goal-grid configuration; P2 only receives information from the working-grid. Each game lasts 6 turns, with P1 taking the first turn. Specifically, on P1's turn, it can move the red block to one of the four cells, remove it from the grid, or pass the turn (similarly for P2 / blue block). Blocks cannot be placed on top of each other. The partners share rewards, and earn 10 points for each correctly placed block, for a maximum of 20 points per game. P1 needs to both place its own red block correctly, and communicate the position of the blue block to P2. As such, success in this task requires a mixture of rule-dependent and convention-dependent skills.

In Figures 9a and 9b we plot the results on adaptation to new hand-designed and self-play partners, where our method outperforms the baselines for non-zero values of $\lambda$. To verify that our task module is learning a good task representation, we compute the distance between the task module output and the oracle marginal (Figure 7). We see that our approach with $\lambda = 0.5$ leads to a smallest gap from the oracle marginals. Finally, similar to the contextual bandit task, we test zero-shot coordination with the same partners on new tasks in Figure 10. We tweak the task by changing the colors of target blocks, and use hand-designed partners to ensure the developed conventions persist. Again, our method performs better on the new task/partner combination than BaselineModular. We provide more details of the experimental setup in the Appendix.

**Hanabi** Finally, we experiment on a light version of the collaborative card game Hanabi, which has been studied in many related works in MARL. We focus on the two-player version of Hanabi, with 1 color and 5 ranks (for a maximum score of 5). Since it is not straightforward how to create a

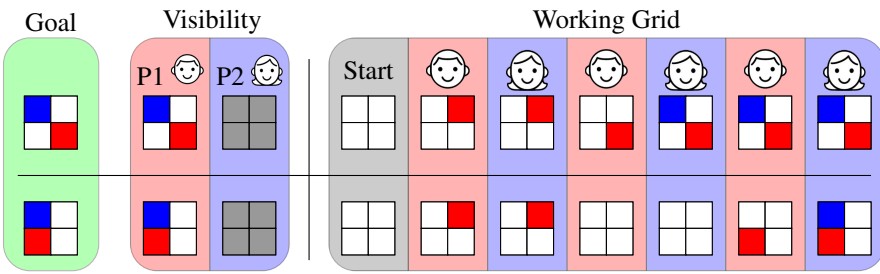

Figure 8: Block placing task: each row displays a new round of the task. On the left, we see the goal-grid and how it appears to each player. Since P2 cannot see the goal-grid, we show a fully grey grid. On the right-side we see the working grid evolve over the course of 6 turns. P1 edits the red block on turns $1, 3, 5$ and P2 edits the blue block on turns $2, 4, 6$.

variety of hand-designed partners, or tweak the task while maintaining the same symmetries, we use self-play partners only and omit experiments on coordinating on a new task with the same partners. The results on adaptation to new self-play partners are shown in Figure 9c.

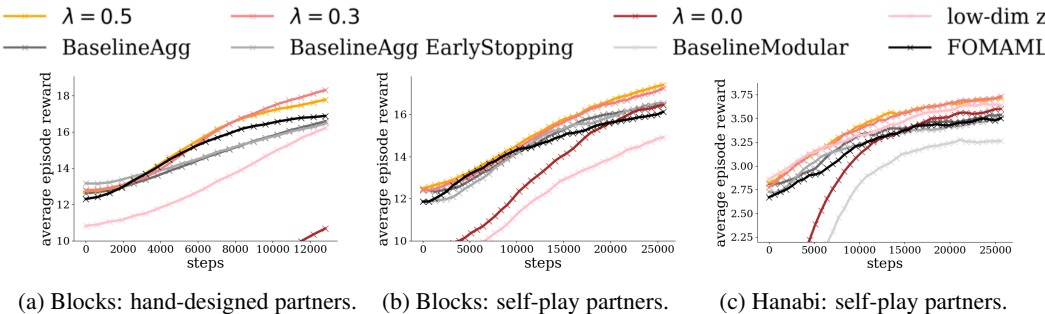

(a) Blocks: hand-designed partners.  (b) Blocks: self-play partners.  (c) Hanabi: self-play partners.

Figure 9: Adapting to a single new partner for block placing task and Hanabi.

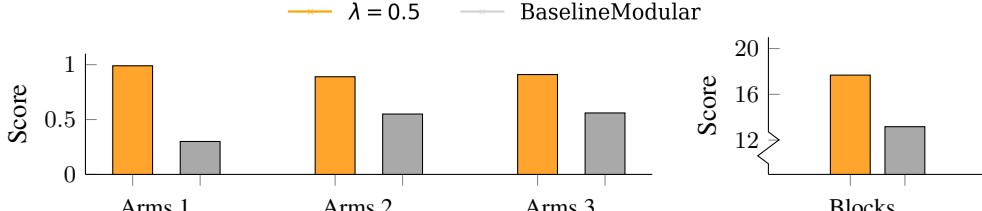

Figure 10: Zero shot performance on new task/partner combination. Higher is better. Orange refers to our method with $\lambda = 0.5$ and grey refers to BaselineModular. Arms $m$ means the task has $m$ contexts with symmetries. We create the new task by altering contexts where there is only one good action (as a result, we omit Arms 4 since it has no contexts with only one good action). We do not compare with BaselineAgg since it is non-modular and cannot be composed for new task/partner combinations.

## 7 CONCLUSION

We study the problem of adapting to new settings in multi-agent interactions. To develop AI-agents that quickly adapt to new partners and tasks, we introduced a framework that learns rule-dependent and convention-dependent representations. Our AI agents can adapt to conventions from new partners without re-learning the full complexities of a task, and can carry over existing conventions with same partners on new tasks. We run a study of human-human interactions that suggests human conventions persist across tasks with similar symmetries. Our experiments show improvements in adapting to new settings on a contextual bandit task, a block placing task, and Hanabi.

ACKNOWLEDGMENTS

This work was supported by NSF (#1941722, #2006388), and the DARPA HICON-LEARN project.

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

# A  PROOF OF LEMMA 1

*Proof.* The marginal of the best-response strategy across partners at state $s$ of a task with reward $R$ can be written as:

$$p(a_{\boldsymbol{e}}) = \mathbb{E}_{\rho \sim \theta_\rho}[f(s, a_{\boldsymbol{e}}, \rho(s, Q_s^R))]$$

When the ego agent is deterministic, its best-response strategy $f(s, a_{\boldsymbol{e}}, a_{\boldsymbol{p}})$ can be rewritten as a function $f'$ where $f'(s, a_{\boldsymbol{p}}) = a_{\boldsymbol{e}}$ if $f(s, a_{\boldsymbol{e}}, a_{\boldsymbol{p}}) = 1$. Then, we can write the likelihood of a best-response strategy from a newly sampled partner as a function of the marginal:

$$\mathbb{P}_{\rho \sim \theta_\rho}[f'(s, \rho(s, Q_s^R)) = a_{\boldsymbol{e}}] = p(a_{\boldsymbol{e}})$$

$\square$

Loosely speaking, when the best-response strategies are not deterministic, storing only their marginals throws away information. But in the case when the strategies are deterministic, each strategy can be represented by a single action, so we are simply showing that storing their marginals does not throw away information.

# B  ARCHITECTURE DETAILS AND HYPERPARAMETERS

The specific size of each module in Figure 4 is shown in Table 1. We train the policy network using Proximal Policy Optimization (PPO) (Schulman et al., 2017) with the Stable Baselines software package (Raffin et al., 2019). For the state-value function, we use a network of the same size, and combine the value predictions from the task module and the partner module by adding them together.

|  | Bandit | Blocks | Hanabi |
|---|---|---|---|
| Task Module Layer Size | 30 | 80 | 500 |
| Partner Module Layer Size | 30 | 80 | 500 |
| Low Dimensional $z$ | 5 | 20 | 50 |

Table 1: Architecture: each module is a 2-layer fully connected MLP with ReLU activation. Values in the table denote the size of each layer. We set the number of latent variables $z$ outputted by the task module to be the same as the layer size in all cases except for the low dimensional $z$ setting, where we use the dimensions reported here.

|  | Bandit | Blocks | Hanabi |
|---|---|---|---|
| Timesteps | 10000 | 1e6 | 5e5 |
| Minibatch size | 16 | 40 | 160 |
| Num. epochs | 20 | 10 | 5 |
| Learning Rate | 3e-4 | 3e-4 | 3e-4 |

Table 2: Hyperparameters for training self-play partners.

|  | Bandit | Blocks | Hanabi |
|---|---|---|---|
| Minibatch size | 16 | 160 | 160 |
| Num. epochs | 50 | 5 | 5 |

Table 3: Hyperparameters for adapting to new partners.

# C  EXPERIMENTAL SETUP

Code for the experiments in our paper is available at https://github.com/Stanford-ILIAD/Conventions-ModularPolicy.

## C.1  CONTEXTUAL BANDIT

**Task setup** We use a contextual bandit with 4 contexts and 8 arms. We label the contexts $1, \ldots, 4$ and the arms $1, \ldots, 8$. In the initial setup, at context $i$ the optimal actions (green boxes) are arms $i$

|  | Bandit | Blocks | Hanabi |
|---|---|---|---|
| Inner iterations | 4 | 4 | 4 |
| Inner learning rate | 3e-4 | 3e-4 | 3e-4 |
| Outer step size | 0.25 | 0.25 | 0.25 |

Table 4: Hyperparameters for First-Order MAML.

and $i + 4$. So, all 4 contexts have symmetries, since there are multiple equally optimal actions. We experiment with different versions of the task by changing the number of contexts with symmetries, from 4 down to 1. To get $m$ contexts with symmetries, we set the optimal action of a context $i$ to only arm $i$, if $i > m$. So, contexts $[1, \ldots, m]$ have two equally optimal actions, and contexts $[m+1, \ldots, 4]$ have only one optimal action. Naturally, when there are fewer contexts with symmetries, coordination is easier.

**Partner setup**  We use both partners obtained by self-play and hand-designed partners. For the self-play partners, we train using PPO with the hyperparameters shown in Table 2. For the hand-designed partners, the partners were designed to pick any one of the equally optimal actions and to stick to the same choice. The distribution over choices across the hand-designed partners was made to be uniform over the optimal actions. We used a set of 10 self-play partners and 4 hand-designed partners for training, and a set of the same size of testing.

**Tweaked task**  For coordinating on a new task (Figure 10), we change the optimal action for contexts where there no symmetries. As such, this is only possible when the number of symmetries $m$ is less than 4. For the new task, a context $i$ where $i > m$, we modify the optimal action from $i$ to $i + 4$. The contexts with symmetries remain unchanged. For the hand-designed partners, we ensured that they stuck to the same choices (conventions) as the initial task in the contexts with symmetries.

## C.2  BLOCK PLACING

**Task setup**  We used the task setup described in Section 6. In our experiments, the ego agent is P1 and the partner agents are P2.

**Partner setup**  We use both partners obtained by self-play and hand-designed partners. For the self-play partners, we train using PPO with the hyperparameters shown in Table 2. For the hand-designed partners, we design the partners to interpret the first action (turn 1) of the ego agent based on a fixed permutation of the block positions. For example, a possible partner may always place the blue block top-right whenever the ego agent places the red block top-left on turn 1. We used a set of 6 self-play partners and 3 hand-designed partners for training, and a set of 6 self-play partners and 6 hand-designed partners for testing.

**Tweaked task**  For coordinating on a new task (Figure 10), we change the reward function by reversing the interpretation of the red and white cells in the goal grid. In other words, P1 has to place the red block on the working grid in one of the two positions that is a white cell on the goal grid. P2's goal is unchanged; P2 still has to place the blue block where the blue block appears in the goal grid, so the relevant symmetries remain unchanged. For the hand-designed partners, we ensured that they stuck to the same permutations (conventions) as the initial task when interpreting the ego agent's action on turn 1.

## C.3  HANABI

**Task setup**  We used the Hanabi Learning Environment package (Bard et al., 2020), with the following configuration: 1 color, 5 ranks, 2 players, hand size 2, 3 information tokens, and 3 life tokens. The maximum score is 5 points.

**Partner setup**  We use only partners obtained by self-play. For the self-play partners, we train using PPO with the hyperparameters shown in Table 2. We used a set of 4 self-play partners for training, and a set of the same size for testing.

### C.4 Additional Plots

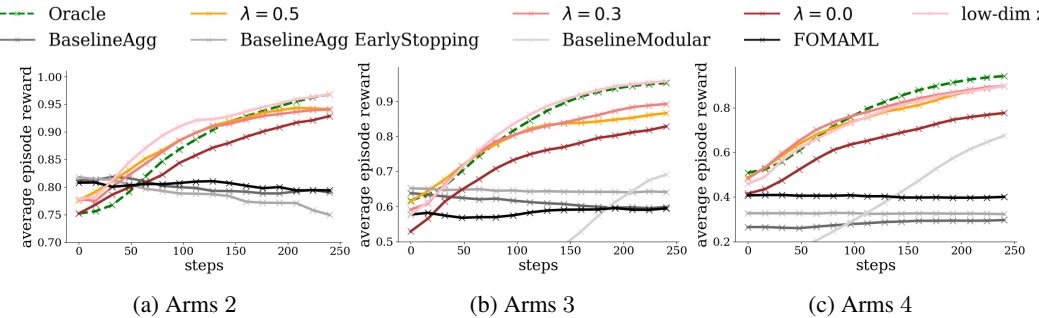

(a) Arms 2                                      (b) Arms 3                                      (c) Arms 4

Figure 11: Contextual bandit game: adapting to a single new partner. Similar to Figure 6, but instead we train and test on self-play partners.

### C.5 User Study Individual Results

Here we provide more detailed results from our user study. For each context and task we report the joint actions made by each pair of users on their first try and on their fifth try.

| ID | Train-A Try 1 | Train-A Try 5 | Test-A Try 1 | Test-A Try 5 | Train-B Try 1 | Train-B Try 5 | Test-B Try 1 | Test-B Try 5 | Train-C Try 1 | Train-C Try 5 | Test-C Try 1 | Test-C Try 5 |
|----|------|------|------|------|------|------|------|------|------|------|------|------|
| 1  | 1/1 | 1/1 | 3/3 | 3/3 | 3/3 | 3/3 | 2/1 | 1/1 | 2/2 | 2/2 | 2/2 | 2/2 |
| 2  | 1/1 | 1/1 | 3/3 | 3/3 | 3/3 | 3/3 | 1/2 | 1/1 | 2/2 | 2/2 | 2/2 | 2/2 |
| 3  | 1/1 | 1/1 | 3/3 | 3/3 | 4/3 | 3/3 | 2/1 | 1/1 | 4/2 | 2/2 | 2/2 | 2/2 |
| 4  | 1/1 | 1/1 | 3/3 | 3/3 | 3/3 | 3/3 | 2/2 | 1/1 | 2/2 | 2/2 | 2/2 | 2/2 |
| 5  | 1/1 | 1/1 | 3/3 | 3/3 | 4/4 | 4/4 | 1/2 | 2/2 | 4/4 | 4/4 | 4/4 | 4/4 |
| 6  | 1/1 | 1/1 | 3/3 | 3/3 | 3/3 | 3/3 | 2/2 | 1/1 | 2/2 | 2/2 | 2/2 | 2/2 |
| 7  | 1/1 | 1/1 | 1/1 | 3/3 | 3/4 | 3/3 | 1/2 | 1/1 | 2/2 | 2/2 | 2/2 | 2/2 |
| 8  | 1/1 | 1/1 | 3/3 | 3/3 | 3/3 | 3/3 | 2/2 | 1/1 | 2/2 | 2/2 | 2/2 | 2/2 |
| 9  | 1/1 | 1/1 | 3/3 | 3/3 | 3/3 | 3/3 | 2/2 | 1/1 | 2/2 | 2/2 | 2/2 | 2/2 |
| 10 | 1/2 | 1/1 | 1/3 | 3/3 | 2/2 | 4/4 | 4/2 | 2/1 | 3/3 | 2/4 | 2/2 | 2/4 |
| 11 | 1/1 | 1/1 | 3/3 | 3/3 | 3/3 | 3/3 | 1/1 | 1/1 | 2/2 | 2/2 | 2/2 | 2/2 |
| 12 | 1/1 | 1/1 | 3/3 | 3/3 | 4/3 | 3/3 | 1/1 | 1/1 | 2/4 | 2/2 | 2/2 | 2/2 |
| 13 | 1/1 | 1/1 | 3/3 | 3/3 | 4/3 | 3/3 | 2/1 | 1/1 | 4/2 | 2/2 | 2/2 | 2/2 |
| 14 | 1/1 | 1/1 | 3/3 | 3/3 | 4/3 | 4/4 | 2/2 | 2/2 | 4/2 | 4/4 | 4/4 | 4/4 |
| 15 | 1/1 | 1/1 | 3/3 | 3/3 | 4/3 | 3/3 | 1/2 | 2/2 | 2/2 | 2/2 | 2/2 | 2/2 |
| 16 | 1/1 | 1/1 | 3/3 | 3/3 | 4/3 | 4/4 | 2/2 | 2/2 | 4/3 | 4/4 | 4/4 | 4/4 |
| 17 | 1/1 | 1/1 | 3/3 | 3/3 | 4/4 | 4/4 | 2/2 | 2/2 | 4/2 | 2/2 | 2/2 | 2/2 |
| 18 | 1/1 | 1/1 | 3/3 | 3/3 | 3/4 | 3/3 | 1/2 | 1/1 | 2/2 | 2/2 | 2/4 | 2/2 |
| 19 | 1/1 | 1/1 | 3/3 | 3/3 | 4/3 | 4/4 | 2/2 | 2/2 | 4/2 | 4/4 | 4/4 | 4/4 |
| 20 | 1/1 | 1/1 | 3/3 | 3/3 | 4/3 | 3/3 | 2/3 | 2/2 | 2/2 | 2/2 | 2/2 | 2/2 |
| 21 | 1/1 | 1/1 | 3/3 | 3/3 | 3/4 | 3/3 | 2/3 | 2/2 | 4/2 | 2/2 | 2/2 | 2/2 |
| 22 | 1/1 | 1/1 | 3/3 | 3/3 | 3/4 | 4/4 | 1/2 | 2/2 | 2/2 | 2/2 | 2/2 | 2/2 |
| 23 | 1/1 | 1/1 | 3/3 | 3/3 | 4/3 | 3/3 | 1/1 | 1/1 | 4/2 | 2/2 | 2/2 | 2/2 |

Table 5: Detailed individual results from the contextual bandit user study. Each cell is of the form $a_1/a_2$ representing the actions made by the two partners. The task is shown in Figure 5.

