# OpenReview forum: "On the Critical Role of Conventions in Adaptive Human-AI Collaboration"
_ICLR.cc/2021/Conference — ICLR 2021 Poster_

### Official Review · AnonReviewer3 · 2020-10-19
**Interesting and sound approach to adaptive behavior in agents (task- and partner-specific). However, it's not clear what the studies' goals are and, as a consequence, what we learn about the object of study.**

**Rating:** 7
**Confidence:** 3

**Review:**

==== On the Critical Role of Conventions in Adaptive Human-AI Collaboration ====
##### Summary #####
This paper studies how artificial agents can be endowed with the human-like capability to, one the one hand, retain behaviors that best fit the task environment(s) when no equally good alternatives exist and, on the other, transfer arbitrary partner-specific conventions to other tasks. Addressing this challenge is important. Chiefly, it promises to improve the number of iterations needed to converge on optimal behavior for cases where analogous strategies were already converged on in a different task / with the same partner. This work proposes to achieve this by combining two separate learned representations. One for partner-specific behaviors. The other for the task itself. The latter module, reused across partners, ends up regulating behavior for cases where only one optimal actions exists. When multiple optimal actions exist there is some slack for players to explore and converge on an arbitrary optimal action. In this sense, agents are endowed with the ability to reuse optimal action policies that will work across agents as well as to reuse optimal partner-specific policies when faced with a context with multiple solutions.

##### Reasons for score #####

I ultimately decided for rejection. This work has many merits: the topic is very important; it is of relevance to many fields; the approach is sound and the experiments interesting. However, I fail to see how much this work advances our understanding of the interplay between partner-specific and task general behavior. The main reason is that, while sound and straightforward to follow, it leaves open a lot of crucial questions (e.g., How do we recognize what a task/context is? Is the separation of the modules motivated? If so, how? What do we learn from the human experiment that we didn't before? See "Cons" below for details). I'm very happy to be convinced otherwise but I don't think that these concerns can be addressed in the present submission.

##### Pros #####
+ Interesting and important topic that is relevant to many fields
+ Technically sound approach and clear exposition.
+ Comprehensive experimentation. I really liked that the approach was put to the test across agents, human and artificial, and tasks (Contexual Bandit, Block Placing; Hanabi)


##### Cons #####

- Goals: My main issue is that the goals of this study are unclear. This work would be greatly enhanced by clarifying what they are and what is ultimately achieved. I do not think its true or fair to state that, as the authors put it in the abstract, "current approaches have not attempted to distinguish a task and the conventions used by a partner, and [that] more generally there has been little focus on leveraging conventions for adapting to new settings". To name just one of many examples, in the linguistics literature this distinction is standardly made and has long been studied (e.g., by Clark & Wilkes-Gibbs, 1986; Clark 1996; Hawkins et al. 2017, all cited in this work). This is also how semantics vs. pragmatics is defined in game-theoretic approaches to language use & dialog (e.g., Franke 2009, "Signal to Act"; or Brochhagen 2017, " Signalling under Uncertainty: Interpretative Alignment without a Common Prior"), as well as how it is generally understood within Gricean pragmatics. In other words, the distinction has been made and rests on a long philosophical tradition. I therefore don't think that taking this separation seriously alone is enough to motivate this investigation.

- Further motivations: In a similar vein to the point above, there's a lack of detail on what sets this work apart from previous investigations. For instance, the critique that "[...]" all these frameworks [...] quickly become intractable" is not very strong in light of the existence of approximations and solutions for the models mentioned (e.g., Monroe 2018's "Learning in the Rational Speech Acts Model", which also uses neural networks to model rule-dependent behavior).

- Clarification: In what sense are Train and Test different tasks? (Section 5)

- Analysis: The experiment in Section 5.1 averages across participants. I'd suggest looking at / reporting individual-level variation. These kinds of experiments usually vary a lot from individual to individual (see, e.g., Kanwal et al. 2017, "Zipf’s Law of Abbreviation and the Principle of Least Effort"). Population-level averages can therefore inadvertently hide or misrepresent what subjects are actually doing.

- On page 4, the authors state: "we assume that behavior at different states are uncorrelated: a choice of action at one state tells us nothing about actions at other states." I take it that this assumption is made for simplicity's sake. However, isn't this also an integral part of human-like abilities? If I realize my partner's behavior accords --or does not accord-- to some conventions I had already established, I might as well behave accordingly. For instance, if I'm playing a game of chess with someone I might match their level of expertise (e.g., avoid castling with a complete beginner); and if I'm speaking with someone at a conference, I might change my phrasing based on theirs based on what I believe to be their background to be.

##### Questions during rebuttal period #####
See cons above

##### Updated reviewer #####
The authors have addressed my main concerns satisfactorily, in a clear and concise matter. I have updated my recommendation to reflect this.

##### Minor comments #####
- Figure 6 has a different y-axis across plots. This is confusing at a fast glance, and makes the subplots' comparison quite hard.

---

> ### Author Response · Authors · 2020-11-13
> **Response to Reviewer 3: recognizing linguistic literature and other thoughtful questions**
>
> **how much this work advances our understanding of the interplay between partner-specific and task general behavior.**
>
> We agree that many works in the linguistics literature have studied the “interplay between partner-specific and task general behavior”. Our work is largely inspired by these works in a language setting, and we are expanding this to games without explicit communication, and to the deep-RL setting. To our knowledge, there has not been as much work considering the implicit communication setting, so we are bringing ideas of conventions from the linguistic literature to coordination games that don’t have an explicit communication channel. Moreover, our focus is on providing a concrete deep-RL approach to learning the two types of partner-specific and task-general behavior.
>
> **How do we recognize what a task/context is?**
>
> In our problem formulation, the ego agent knows the identity of the task (which task it is solving) but does not know the exact reward function of the task. The agent observes the state (context) of the environment at each step. This is typically the setup used in similar problem settings such as meta-learning or multi-task learning. The bigger question of how to recognize when the task has changed and what constitutes a task in the real world is very interesting, and we would be curious to hear thoughts you may have on this.
>
> **Is the separation of the modules motivated? If so, how?**
>
> Intuitively, the task/rules determine the set of equally optimal solutions, and each partner separately chooses which solution based on the developed conventions. We consider this explicit factorization of tasks and partners in the problem setup in Section 4, so we think that the problem of adapting to new partner and task combinations in this setting naturally motivates the separation of modules.
>
> We also believe the partner module should be “on top” of the task module for the following reasoning (copied from our response to Reviewer 1):
> We want the task module to ‘featurize’ the raw observations into latent space z, so that the partner module has an easier learning task: for example, loosely speaking, the partner module only has to choose between equally optimal actions instead of all possible actions. For the same reason, we also think it makes sense to have the modules in this order (instead of task module on top), since we want to first extract the optimal actions, and then second have the partners choose one of the optimal actions second.
>
> **What do we learn from the human experiment that we didn't before?**
>
> Our intuition that coordinating with a partner on an earlier task will help us coordinate with the same partner on a new task relies “on the hypothesis that our partners will carry over the same conventions developed on an earlier task” (section 5.1). Since the users had no external communication, it is not clear to what extent this hypothesis holds true. Our human experiments show that when the Q-value is exactly the same (in context C of Train/Test), they are generally able to carry over conventions. When the Q-values are different in context B of Train/Test, but they could conceivably have mapped Train action 3 to Test action 1 and Train action 4 to Test action 2 and carried over the conventions, but our users were generally not able to carry over their convention in this way.
>
> **the goals of this study are unclear**
>
> The goal of our work is to provide a concrete “learning framework that teases apart rule-dependent representation from convention-dependent representation in a principled way” (from the abstract) to enable faster adaptation to new tasks and partners. We focus on this goal throughout the paper, in the environment setup (section 4) where there is a factorization of partner and task, the architecture (section 5) that separates representations with modular networks, and in the human study where we test if the hypothesis of carrying over convention-dependent representation is true. In the experiments we test on a variety of tasks and environments that require coordination with implicit communication, where we show how our method performs when adapting to new partners and tasks. We hope this helps clarify the goal of our work.

---

> > ### Author Response · Authors · 2020-11-13
> > **Response to Reviewer 3: continued**
> >
> >
> > **in the linguistics literature this distinction is standardly made and has long been studied...and rests on a long philosophical tradition**
> >
> > Yes, we agree that the works in the linguistic literature have studied this distinction, and in fact many of these works are what inspired this project. We thank you for the references, and we have referred to these works in the updated Introduction section. We are happy to include other relevant works from the linguistic literature.
> >
> > That being said, to the best of our knowledge many of the works cited do not provide a systematic learning-based approach that leverages this distinction of conventions. The goal of our work is to go beyond recognizing that conventions exist, and to concretely obtain a latent representation of the conventions that we can transfer over when learning in new settings. Additionally, we focus on the tasks with implicit communication, which has been less-studied in previous works.
> >
> > **there's a lack of detail on what sets this work apart from previous investigations. For instance, the critique that "[...]" all these frameworks [...] quickly become intractable" is not very strong in light of the existence of approximations and solutions for the models mentioned (e.g., Monroe 2018's "Learning in the Rational Speech Acts Model", which also uses neural networks to model rule-dependent behavior).**
> >
> > The fundamental difference is that we are proposing to learn representations of the tasks and partners and to fine-tune them to adapt to new settings, rather than approximating a belief modeling approach as in Monroe 2018. We believe that full belief modeling may not be the only (or even the right) perspective on this problem. Specifically we are skeptical if belief modeling or even approximations of it is what is actually done in certain human-human interactions, so motivated by this different perspective, we propose to learn representations of the task/partner, which seems more natural for some coordination problems. (It would be interesting to consider which coordination problems are more suitable for belief modeling (perhaps chess) vs learning representations (perhaps Hanabi)).
> >
> > Thank you for this suggested reference: we have included it in the PDF and elaborated on this difference in the Related Work section.
> >
> > **Clarification: In what sense are Train and Test different tasks? (Section 5)**
> >
> > The location of the green cells have shifted, so the reward function is now different. For example in context B, the partners must pick one of the left two cells in the Train task, and they must pick one of the right two cells in the Test task. As we describe in Section 4, different tasks differ in their reward functions, but keep the same ‘domain” (state/action/transitions).
> >
> > **Analysis: The experiment in Section 5.1 averages across participants. I'd suggest looking at / reporting individual-level variation. These kinds of experiments usually vary a lot from individual to individual (see, e.g., Kanwal et al. 2017, "Zipf’s Law of Abbreviation and the Principle of Least Effort"). Population-level averages can therefore inadvertently hide or misrepresent what subjects are actually doing.**
> >
> > Thanks for bringing this to our attention. We completely agree and from looking at our data, the individual variation seems comparable to aggregate results. We are happy to share details on individual-level variations, and we have added more detailed individual data in the Appendix (Table 5).
> >
> > **On page 4, the authors state: "we assume that behavior at different states are uncorrelated: a choice of action at one state tells us nothing about actions at other states." I take it that this assumption is made for simplicity's sake. However, isn't this also an integral part of human-like abilities? If I realize my partner's behavior accords --or does not accord-- to some conventions I had already established, I might as well behave accordingly. For instance, if I'm playing a game of chess with someone I might match their level of expertise (e.g., avoid castling with a complete beginner); and if I'm speaking with someone at a conference, I might change my phrasing based on theirs based on what I believe to be their background to be.**
> >
> > Indeed we made the assumption for simplicity’s sake. We do agree that humans may predict conventions in one state based on conventions in another state (going along with your example, if your partner has castled, it is more likely your partner knows how to take en passant).  This is definitely an interesting future step to look into, although it may require some domain knowledge or a much larger number of partners to accurately model the relationship between conventions in different states.
> >
> > **Figure 6 has a different y-axis across plots. This is confusing at a fast glance, and makes the subplots' comparison quite hard.**
> >
> > Thanks, we have fixed this.

---

### Official Review · AnonReviewer1 · 2020-10-26
**Good paper on an important topic**

**Rating:** 7
**Confidence:** 4

**Review:**

This paper proposes that in human-AI collaboration using deep neural nets, the AI agents we train should separate learning the _rules_ of the environment from the _conventions_ used to coordinate with humans in that environment. It proposes a simple method to do so: learn a single task-specific module that is always used, as well as many partner-specific modules that are used with specific partners. Intuitively, when trained with multiple partners, the task-specific module should learn heuristics that work across partners (the environment-specific rules) while the partner-specific model should learn personalized heuristics for each partner (the conventions). They further incentivize this by regularizing the task-specific module towards the average of the partner policies.

Quality: The one qualm I have is that the environments studied are relatively simple (though even 1-color Hanabi is a fairly challenging coordination problem). Other than that the paper is high quality. I especially appreciated the user study.

Clarity: I found the paper reasonably clear, though some parts took some time to understand.

Originality: To my knowledge, this is the first paper exploring the distinction between rules and conventions within the deep RL paradigm, and it shows good results both in simulation and with real humans in a user study (albeit in a very simple toy domain).

Significance: Human-AI collaboration is clearly important and significant, and the application of deep RL to human-AI collaboration has grown in the last 2-3 years. This paper extends this field with an important contribution.

The main weakness of the approach I see is that it doesn’t have an obvious way to handle the fact that humans will typically _adapt_ to whatever policy the robot plays. This may not happen in the simple environments considered in this paper, but definitely does happen in larger environments. Perhaps this technique would work anyway: arguably, an adaptive human just means that the convention changes, and simply continuing to train the partner module could be enough for the robot to adapt to this change in the human’s convention.

Regardless, I think even the contribution of how to deal with multiple different non-adaptive humans is significant and relevant to ICLR.

Questions for the authors:

1. Why does the partner module operate “on top of” the task module? Why not instead have both modules take in the state as an input and produce a probability distribution over actions, that are then multiplied together?
2. How might this extend to collaboration with adaptive humans?

Typos / nitpicks:

The phrase “ego agent” was confusing to me, and I think it wasn’t explained anywhere? I did eventually figure it out though.

---

> ### Author Response · Authors · 2020-11-13
> **Response to Reviewer 1: clarifications and on adaptive humans**
>
> **The one qualm I have is that the environments studied are relatively simple (though even 1-color Hanabi is a fairly challenging coordination problem). Other than that the paper is high quality. I especially appreciated the user study.**
>
> We agree that it would be interesting to try on even more environments. However we would like to emphasize that “even 1-color Hanabi is a fairly challenging coordination problem”. The block placing task is also challenging too -- it’s difficult to coordinate when there’s no external communication.
>
> **The main weakness of the approach I see is that it doesn’t have an obvious way to handle the fact that humans will typically adapt to whatever policy the robot plays....perhaps this technique would work anyway...**
>
> Yes, we currently do not focus on humans adapting to the robot policy. This would be interesting to study, but since training with human-in-the-loop is too costly, a big question is how to realistically model human adaptation. We think this is an intriguing next step for future work. We are currently looking into studying the problem of conventions with adaptive AI partners (where data collection is not a problem), and we're working on developing techniques that could be applied in such settings with the goal of moving to adaptive human-AI interaction in the future. We also agree that a modular architecture may also be the right approach for adaptive partners too.
>
> **Why does the partner module operate “on top of” the task module? Why not instead have both modules take in the state as an input and produce a probability distribution over actions, that are then multiplied together?**
>
> We want the task module to ‘featurize’ the raw observations into latent space z, so that the partner module has an easier learning task: for example, loosely speaking, the partner module only has to choose between equally optimal actions instead of all possible actions. For the same reason, we also think it makes sense to have the modules in this order (instead of task module on top), since we want to first extract the optimal actions, and then second have the partners choose one of the optimal actions second.
>
> **How might this extend to collaboration with adaptive humans?**
>
> Perhaps fine-tuning the partner module to adapt to adaptive partners would be a potential approach. We would need a good grasp on how we think humans adapt, which is an interesting question on its own.
>
> **The phrase “ego agent” was confusing to me, and I think it wasn’t explained anywhere? I did eventually figure it out though.**
>
> Thank you for bringing this up. We have clarified it in Section 3.

---

> > ### Comment · AnonReviewer1 · 2020-11-25
> > **Thanks for the response!**
> >
> > Thanks for the clarifications! (And sorry for the late response.)
> >
> > > We want the task module to ‘featurize’ the raw observations into latent space z, so that the partner module has an easier learning task: for example, loosely speaking, the partner module only has to choose between equally optimal actions instead of all possible actions. For the same reason, we also think it makes sense to have the modules in this order (instead of task module on top), since we want to first extract the optimal actions, and then second have the partners choose one of the optimal actions second.
> >
> > This makes sense, but it enforces a particular structure on conventions: they must be some transformation of the optimal actions. My guess would be that results would improve if you did share information between $g^t$ and $g^p$, but without requiring this particular structure. For example, you could have a shared feature encoder $f$, so that you compute latent representations $z = f(s)$, and then $g^t$ and $g^p$ could be (different) linear functions of $z$ (or they could be 2-layer MLPs on top of $z$, if more depth is required).
> >
> > In any case, this is a minor point and not one that particularly needs to be addressed.

---

> > > ### Author Response · Authors · 2020-11-25
> > > **Interesting suggestion**
> > >
> > > > a shared feature encoder $f$, so that you compute latent representations $z=f(s)$, and then $g^t$ and $g^p$ could be (different) linear functions of $z$ (or they could be 2-layer MLPs on top of $z$, if more depth is required)
> > >
> > > That is an interesting suggestion! Our current structure is indeed motivated by considering conventions as some sort of "transformation of the optimal actions" (i.e. the model in Section 4). Exploring other structures (like the ones you suggest) would definitely be a worthy direction to try out.

---

### Official Review · AnonReviewer2 · 2020-10-29

**Rating:** 7
**Confidence:** 4

**Review:**

This paper makes the observation that when performing cooperative tasks with partners, there are two components to learn: how to perform the task, and how to coordinate with the partner according to conventions. Therefore, it proposes to separate these two components via a modular architecture, which learns a task-specific action distribution that attempts to marginalize out all possible partner conventions, and a series of partner-specific action distributions. The agent's own policy is the product of these two distributions. When coordinating with a new partner, the partner-specific component is learned from scratch using a pre-trained task-specific component, and vice versa.

The paper goes after an ambitious and useful problem (rapid adaptation to coordinating with novel partners on new tasks), and proposes a novel technique for doing so. A weakness is that the paper does not use reasonable baselines, and effectively compares only to ablations of their own model. Why not compare to a meta-learning technique? Or compare to some of the existing SOTA methods for Hanabi?

In general the paper is well written, but it could be made significantly clearer by providing further details on how the partner action distributions g^p_i(a|s) are obtained. Given the explanation in the beginning of Section 4, I was initially under the impression that these represented the partner's policy distribution produced by its Q-values, or perhaps the partner's actual action frequencies obtained from observing trajectories. However, it seems that the model is learned entirely end-to-end, and so these distributions actually represent how the agent's own policy should be modified according to which partner it is playing with. Is this correct? If so, this explanation should be added to the paper to make it more clear how the technique can apply beyond simple domains like the contextual bandit, in which agents must choose the *same* actions as the partner.

The fact that the partner module must be re-initialized and learned from scratch for each new partner is a weakness of the method.  Why not learn some type of partner embedding that would enable generalization to new partners at test time that use similar conventions to training partners?

The experiments section of the paper felt rushed and lacking in explanation compared to the first 6 pages. The clarity/impact could be enhanced by explaining the experiments in more detail. In particular, the block placing task is not explained (do agents place blocks separately? do they have to place a block together at the same time?). Also, the need for "hand-designed" partners is not explained, nor is what they are hand-designed to do.

Since the paper collects a human user study on conventions, why not test how well the trained models are able to coordinate with humans? This would significantly enhance the impact of the paper.

Other suggestions:
- Figure 2 caption does not include the explanation that agents must choose the same action to get a reward
- A legend should be added to Figures 7, 8, and 9.
- Figure 7 is interesting in that even without the Wasserstein distance penalty (when lambda=0), the Wasserstein distance to the ground truth marginal best response is still low, suggesting the model is learning some level of task-specific representation just due to the architecture. This could be explained further in the text.

Edit: I have updated my score based on the new experiments added during the rebuttal process.

---

> ### Author Response · Authors · 2020-11-13
> **Response to Reviewer 2: new experiments and baselines**
>
> **A weakness is that the paper does not use reasonable baselines, and effectively compares only to ablations of their own model. Why not compare to a meta-learning technique? Or compare to some of the existing SOTA methods for Hanabi?**
>
> Thanks for the suggestion. We looked at first-order MAML (FOMAML) / Reptile, and have added them as baselines for comparison. We used the hyperparameters from [On First-Order Meta-Learning Algorithms], with k=4 steps in each inner-loop and step-size 0.25, where they showed that FOMAML has similar performance to MAML in the supervised learning setting. We added the results to the plots in Experiments, and the hyperparameters to the Appendix. In general FOMAML seem to be comparable to BaselineAgg (which is equivalent to using k=1 steps in each inner loop). In some of the plots (e.g. 6a and 6d) it appears to start off strong but slightly worsen after many steps, perhaps because it is optimizing to only “look ahead” a few (k=4) number of steps.
>
> We are not aware of existing methods that target adaptation to new partners in Hanabi. Current SOTA approaches for Hanabi (e.g. Hanabi-SAD) rely on centralized training, which is not applicable to our setting.
>
> **In general the paper is well written, but it could be made significantly clearer by providing further details on how the partner action distributions g^p_i(a|s) are obtained.... these distributions actually represent how the agent's own policy should be modified according to which partner it is playing with. Is this correct? If so, this explanation should be added to the paper to make it more clear how the technique can apply beyond simple domains like the contextual bandit, in which agents must choose the same actions as the partner.**
>
> That is correct. Section 4 describes how the partners actions are determined. Roughly, a partner in our setting can be characterized as a tie-breaking function at states with symmetries (as determined by the Q values).
>
> The g^p_i(a|s), on the other hand, is indeed “how the agent’s own policy should be modified according to which partner it is playing with” (i.e. how should the ego agent respond to the partner). As you point out, in the contextual bandit task, symmetries in the partner’s action space are equivalent to symmetries in the ego agent’s action space, but this is not true in general (e.g. block game or hanabi). The distributions g^p_i(a|s) is targeting the latter case of breaking symmetries from the perspective of the ego agent’s action space, so our approach can handle blocks/hanabi tasks where the symmetries are not “mirrored”.
>
> **The fact that the partner module must be re-initialized and learned from scratch for each new partner is a weakness of the method. Why not learn some type of partner embedding that would enable generalization to new partners at test time that use similar conventions to training partners?**
>
> We did have this idea in mind, which is why the partner modules take in the latents z as input instead of the state s directly, so that the task module can pass along only the features that are relevant to different conventions. We did not try more sophisticated methods of learning the partner embedding in the original version.
>
> Based on your suggestion, we added new experiments where we bottleneck the dimensionality of z to encourage the task module to learn better embeddings for “generalization to new partners at test time”. We think this is perhaps the simplest and more natural form of embedding, and that perhaps the range of human interactions can be modeled in low-dimensional space. We’ve added new curves for this low-dimensional z setting to Figure 6 and 9. But, in general using the low-dimensional z on top of the task module regularization did not seem to do better than the task module regularization alone. It is likely that we need to impose additional structure in the latent space to learn a useful embedding, and this would be an interesting topic for future research.

---

> > ### Author Response · Authors · 2020-11-13
> > **Response to Reviewer 2: continued**
> >
> > **The experiments section of the paper felt rushed and lacking in explanation compared to the first 6 pages. The clarity/impact could be enhanced by explaining the experiments in more detail. In particular, the block placing task is not explained (do agents place blocks separately? do they have to place a block together at the same time?). Also, the need for "hand-designed" partners is not explained, nor is what they are hand-designed to do.**
> >
> > We apologize that this section is not as clear. The exact details of the block placing task were placed in the Appendix due to space limitations, but we have moved it up to the main paper. The agents place blocks separately (turn-based), each player can only move the block of his/her own color, and blocks cannot be placed on top of each other. The high-level goal is that Player2 does not see the goal-grid, so must develop a signaling convention with Player1 in order to interpret hints from Player1 and solve the task.
> >
> > The hand-designed partners were designed to have a diverse range of conventions. We included this set of experiments in case the set of partners generated by self-play were too similar. We wanted a diverse range of conventions to check if our model is actually adapting to new conventions (rather than just memorizing the same convention).
> >
> > **Since the paper collects a human user study on conventions, why not test how well the trained models are able to coordinate with humans? This would significantly enhance the impact of the paper.**
> >
> > That’s a great suggestion. We have added an additional plot (Figure 6d) for the contextual bandit task where we use the data from the human user study in place of hand-designed partners. We observe similar trends (modular architecture works best, and regularization helps), and we agree that testing this setting enhances the impact of the paper.
> >
> > Directly pairing our models with humans would be even more ideal, however  the amount of data currently required to adapt to a human partner is too high to make it practical. There is a need for developing data-efficient techniques that can train with humans in the loop, and this is an interesting topic of future research (e.g. incorporating active learning).
> >
> > **Other suggestions: Figure 2 caption; Figures 7, 8, and 9 legend; Figure 7 is interesting even without the Wasserstein distance**
> >
> > Thanks for these suggestions! We’ve added them to the updated PDF.

---

> > > ### Comment · AnonReviewer2 · 2020-11-25
> > > **Impressed with the new experiments**
> > >
> > > Thank you for the detailed response! I'm impressed you were able to add the FOMAML baseline and experiment with the human-proxy in the short time frame. These results significantly improve the paper in my opinion, especially because they show the benefit of the technique above generic meta-learning.
> > >
> > > > If so, this explanation should be added to the paper to make it more clear how the technique can apply beyond simple domains like the contextual bandit, in which agents must choose the same actions as the partner.
> > >
> > > Thank you for clarifying this point, however I meant to indicate that other readers of the paper might not pick up on this due to the way the paper is written. Would it be possible to add some of the clarifying explanation you provided in the response to the paper itself?

---

> > > > ### Author Response · Authors · 2020-11-25
> > > > **Thanks**
> > > >
> > > > > These results significantly improve the paper in my opinion
> > > >
> > > > Thanks, glad to hear that. If you don't mind updating the score of your review to reflect this, we would really appreciate it!
> > > >
> > > > > Would it be possible to add some of the clarifying explanation you provided in the response to the paper itself?
> > > >
> > > > Yes, certainly. We will clarify this (since the rebuttal phase is about to end, we will make the changes in the camera-ready).

---

### Official Review · AnonReviewer4 · 2020-11-06
**Interesting Problem - Can benefit from a more clear writing**

**Rating:** 6
**Confidence:** 2

**Review:**

The paper proposes an interesting model to study multi-agent interactions, in uncertain environments. In s nutshell, the model proposed consists of a MDP (Finite state, action, horizon) and two players playing simultaneously (the turn-by-turn model can be subsumed by the simultaneous move model as stated in the paper). In the absence of learning, the finite MDP has an optimal solution. The key contribution of the paper is to focus on instances when the optimal solution is not unique. In a two-player model, this requires symmetry breaking in order for a sample path of the MDP to track the optimal trajectory.

The above is when the MDP and rewards are all well known. The setting in the paper concerns a learning situation where some or all of the components of the MDP is unknown. In this case, the agents must learn the MDP, while breaking symmetry (coordinate) in converging on a sample path closest to an optimal one. The problem, is very interesting and the authors propose a nice formulation to study these questions.

I have a few high-level suggestions to the authors.

1. The model description is mathematically imprecise. Are the agents aiming to optimize the total reward collected, in presence of unknown model and partner ? What information about the partners are known to the agent ? (Is the agent distribution common information ? ) Are the partners assumed to know the underlying MDP, or they are also simultaneously learning ? If the partner are also simultaneously learning, is their "state of knowledge" at the beginning known to the ego agent ?

I understand that having a robust solution to all of the above problems is perhaps too hard. Nevertheless, clarifying the precise setup mathematically will greatly aid the reader.

2. The human experiment, were the users able to communicate to each other in any way ? Were they total strangers to each other, or known acquaintances ? Clearly specifying the "conditions of the experiment", can help take the results in context.

Overall, I believe the paper is attempting to study an interesting (and hard) problem. But my (low) rating is based on the clarity of the presentation.

---

> ### Author Response · Authors · 2020-11-13
> **Response to Reviewer 4: clarifications**
>
> **Are the agents aiming to optimize the total reward collected, in presence of unknown model and partner? What information about the partners are known to the agent ? (Is the agent distribution common information ? )**
>
> Our agent is optimizing the total reward, and knows the identity of the task and partner. Our agent does not know the exact MDP of the task or the policy of the partner, but collects experience with (same task / different partner) or (different task / same partner) combinations during training time. The agent distribution is not explicitly known but we have access to samples (i.e. training partners), and testing partners are drawn (i.i.d) from the same distribution. We have clarified this in Section 5.
>
> **Are the partners assumed to know the underlying MDP, or they are also simultaneously learning ? If the partner are also simultaneously learning, is their "state of knowledge" at the beginning known to the ego agent ?**
>
> No, the partners are not also simultaneously learning, they are assumed to have converged to a fixed convention. The partners do not know the underlying MDP when learning to converge to a convention but have access to samples from the environment / can explore the environment. The ‘state of knowledge’ of the partners is not known to the ego agent; the ego agent only observes the state, joint action, and rewards at the end of each episode.
>
> **The human experiment, were the users able to communicate to each other in any way ? Were they total strangers to each other, or known acquaintances ? Clearly specifying the "conditions of the experiment", can help take the results in context.**
>
> The users were not able to communicate to each other in any way. The only signal they received was the revelation of their partner’s actions and the reward at the end of each try. The users were not total strangers -- they were typically pairs of students or staff.
> Some more details: the experiment was conducted using an online interface. Our users were placed in different rooms before we gave them the task instructions, so there was no way to coordinate “outside” of the game. We have updated the PDF in Section 5.1 to clarify this.

---

### Author Response · Authors · 2020-11-13
**Response to all: thank you for your reviews, we have updated the PDF**

Thank you all for your reviews.

In general, we are happy to hear that the problem we study is “very interesting” and “ambitious”, while being “very important”, and “relevant to many fields”. We appreciate the recognition that we have a “nice formulation” and a “novel technique” (“first paper exploring the distinction between rules and conventions within the deep RL paradigm”), and that our paper is an “important contribution” that is “significant and relevant to ICLR”. We also acknowledge that we have “comprehensive experimentation” and show “good results both in simulation and with real humans", with a “technically sound approach and clear exposition”.

We have responded to each of the reviews separately, and we have updated the PDF to reflect the changes as discussed. We thank you for your insightful comments, and we hope we have clarified some of the points brought up in your feedback, and look forward to continuing the discussion.

---

### Decision · Program_Chairs · 2021-01-07
**Final Decision**

**Decision:**

Accept (Poster)

**Comment:**

This paper proposes a new paradigm for learning to perform cooperative tasks with partners, which factors the problem into two components: how to perform the task and how to coordinate with the partner according to conventions. The setting is new and the reviewers are excited about the paper. A clear accept.